# A 2.8 Å Structure of Zoliflodacin in a DNA Cleavage Complex with *Staphylococcus aureus* DNA Gyrase

**DOI:** 10.3390/ijms24021634

**Published:** 2023-01-13

**Authors:** Harry Morgan, Magdalena Lipka-Lloyd, Anna J. Warren, Naomi Hughes, John Holmes, Nicolas P. Burton, Eshwar Mahenthiralingam, Ben D. Bax

**Affiliations:** 1Medicines Discovery Institute, Cardiff University, Cardiff CF10 3AT, UK; 2Diamond Light Source, Harwell Science and Innovation Campus, Didcot OX11 0DE, UK; 3Cardiff School of Biosciences, Cardiff University, Cardiff CF10 3AX, UK; 4Inspiralis Limited, Norwich Research Park Innovation Centre, Norwich NR4 7GJ, UK

**Keywords:** zoliflodacin, quinolones, DNA gyrase, topoisomerase IV, ESKAPE, antibiotic, spiropyrimidinetrione, NBTI, gepotidacin, structure

## Abstract

Since 2000, some thirteen quinolones and fluoroquinolones have been developed and have come to market. The quinolones, one of the most successful classes of antibacterial drugs, stabilize DNA cleavage complexes with DNA gyrase and topoisomerase IV (topo IV), the two bacterial type IIA topoisomerases. The dual targeting of gyrase and topo IV helps decrease the likelihood of resistance developing. Here, we report on a 2.8 Å X-ray crystal structure, which shows that zoliflodacin, a spiropyrimidinetrione antibiotic, binds in the same DNA cleavage site(s) as quinolones, sterically blocking DNA religation. The structure shows that zoliflodacin interacts with highly conserved residues on GyrB (and does not use the quinolone water–metal ion bridge to GyrA), suggesting it may be more difficult for bacteria to develop target mediated resistance. We show that zoliflodacin has an MIC of 4 µg/mL against *Acinetobacter baumannii* (*A. baumannii*), an improvement of four-fold over its progenitor QPT-1. The current phase III clinical trial of zoliflodacin for gonorrhea is due to be read out in 2023. Zoliflodacin, together with the unrelated novel bacterial topoisomerase inhibitor gepotidacin, is likely to become the first entirely novel chemical entities approved against Gram-negative bacteria in the 21st century. Zoliflodacin may also become the progenitor of a new safer class of antibacterial drugs against other problematic Gram-negative bacteria.

## 1. Introduction

Zoliflodacin is an oral spiropyrimidinetrione antibiotic currently in a phase III clinical trial for the treatment of gonorrhea, a sexually transmitted infection (STI) caused by the Gram-negative bacteria *Neisseria gonorrhoeae* (*N. gonorrhoeae*) [1,2,3,4]. Zoliflodacin (Figure 1a) was developed from QPT-1 (or PNU-286607), a compound discovered in Pharmacia via whole-cell screening against Gram-negative (and Gram-positive) bacteria [5]. QPT-1 (Figure 1b), discovered for its antibacterial whole-cell activity, was found to inhibit the bacterial type IIA topoisomerases, *Escherichia coli* (*E. coli)* DNA gyrase (IC_50_ 9 µM) and *E. coli* topo IV (IC_50_ 30 µM) [5]. This method of discovery is reminiscent of the discovery of quinolone and fluoroquinolone antibiotics, which were also initially discovered for whole-cell activity and then found to be inhibitors of the bacterial type IIA topoisomerases [6].

Topoisomerases are essential enzymes needed to relieve topological problems when the DNA double helix is unwound for both DNA replication and transcription [7]. Topoisomerases are divided into type I topoisomerases, which introduce single-stranded DNA breaks to modify the DNA topology, and type II topoisomerases, which modify the topology by introducing double-stranded DNA breaks [7,8,9]. Most bacteria possess two type IIA topoisomerases, DNA gyrase and topo IV. While DNA gyrase can uniquely introduce negative supercoils into DNA, topo IV has good decatenase activity [9,10]. A mechanism for topological changes introduced by DNA gyrase is shown in Figure 1.

The introduction of double-stranded breaks into DNA is potentially hazardous for the cell, and the stabilization of DNA cleavage complexes by quinolones is often bactericidal [11,12].

The proposal that Gram-negative bacteria evolved a second cell wall to protect them from antibiotics produced by other micro-organisms [15] may partly explain the failure of new classes of antibiotics targeting Gram-negative bacteria to date in the 21st century [16,17,18]. Perhaps for Gram-negative bacteria the hardest task is to get antibiotics into the cells, and a whole-cell screening approach followed by the target identification of proven targets is more likely to be successful [19,20]. Indeed, GlaxoSmithKline discovered and developed the NBTI gepotidacin, another new class of DNA-gyrase-targeting antibiotics currently in phase III clinical trials [21], from a hit compound active in a screen for whole-cell antibacterial activity [13]. The chemical diversity of NBTIs such as gepotidacin, which stabilize single-stranded DNA cleavage complexes with bacterial type IIA topoisomerases, suggested that this class of compounds could not have a chemistry-based name [13,22,23,24,25,26,27]. The name NBTI, although originally a pneumonic for novel bacterial topoisomerase inhibitor [13], could also be taken to stand for non-DNA cleavage pocket binding on the two-fold axis inhibitor (as this describes the binding mode of the chemically diverse NBTIs [13,22,23,24,25,26,27]).

The occurrence of antimicrobial resistance in hospital-acquired ESKAPE pathogens (*Enterococcus faecium*, *Staphylococcus aureus*, *Klebsiella pneumoniae*, *Acinetobacter baumannii*, *Pseudomonas aeruginosa*, and *Enterobacter* species) was a major cause for concern in 2009 [28]. New classes of antibiotics have now been developed for Gram-positive bacteria, such as the tiacumicin Fidaxomicin for *Clostridioides difficile* [29]. However, Gram-negative bacteria (the KAPE in ESKAPE) remain a major cause for concern. The popular quinolone and fluoroquinolone antibacterial agents were discovered over sixty years ago from a whole-cell screening approach against Gram-negative bacteria [6]. Since then, the field of chemistry has expanded the quinolone activity to include such agents as delafloxacin, approved in 2017 for treating acute bacterial skin infections caused by the Gram-positive *S. aureus*. Some thirteen out of thirty-eight new antibiotics introduced between 2000 and 2019 were quinolones [16,17,18,30]. However, safety concerns about quinolone side effects have prompted regulatory recommendations to limit the use of quinolones to patients who do not have other treatment options in both Europe (https://www.ema.europa.eu/en/medicines/human/referrals/quinolone-fluoroquinolone-containing-medicinal-products, accessed on 4 January 2023) and the USA (https://www.fda.gov/drugs/drug-safety-and-availability/fda-drug-safety-communication-fda-advises-restricting-fluoroquinolone-antibiotic-use-certain, accessed on 4 January 2023).

The determination of the specific DNA sequences cleaved by DNA gyrase or topo IV [31,32] was important in determining the structures of quinolones in DNA cleavage complexes. In particular, structural studies showed that quinolone antibiotics stabilize double-stranded DNA cleavage complexes with the two bacterial topoisomerases, topo IV [33] and DNA gyrase [34,35], by interacting with ParC or GyrA via a water–metal ion bridge [12,33]. Although the original DNA sequences used in two papers describing structures showing the water–metal ion bridge [33,35] were defined in 2005 [32] and were initially used in structures with *S. pneumoniae* topo IV [36,37], they are asymmetric, and in high enough resolution structures the DNA was clearly averaged around the two-fold axis of the complex [35]. In this paper we used a two-fold symmetric 20-mer DNA duplex to avoid such problems [34,38]. This 20-mer homoduplex DNA was previously used in determining structures with the progenitor of zoliflodacin, QPT-1 [34].

Herein, we describe a 2.8 Å X-ray crystal structure of zoliflodacin in a DNA cleavage complex with *S. aureus* DNA gyrase. The structure is compared with a structure with the quinolone moxifloxacin, also in a DNA cleavage complex with *S. aureus* DNA gyrase. We also show that zoliflodacin has reasonable activity against *A. baumannii* (MICs of 4 µg/mL; the A in ESKAPE). In 2018, a World Health Organization (WHO) priority list [39] proposed developing new drugs active against multidrug-resistant tuberculosis and Gram-negative bacteria. While spiropyrimidinetriones related to zoliflodacin are being developed against *M. tuberculosis* [40,41], the WHO critical priority, carbapenem-resistant *A. baumannii* [39], still urgently requires the development of new antibiotics [42].

The DNA cleavage gate of bacterial type IIA topoisomerases, when complexed with DNA and compounds, seems inherently flexible and usually gives low- or medium-low-resolution data [13,14,37] (we define low (>3 Å), medium-low (2.5–2.99 Å), medium-high (2.01–2.49 Å), and high (<2 Å) based on the confidence in determining the water structures around ligands and metal ions). Deleting the Greek key domain from a *S. aureus* GyrBA fusion truncate allowed the resolution of a complex with GSK299423 and DNA to be improved from 3.5Å to 2.1 Å [13], and this *S. aureus* DNA gyrase fusion truncate (Figure 1e) construct (used in this paper) has given the only high-resolution structures of DNA complexes of bacterial type IIA topoisomerases obtained to date [10].

## 2. Results

### 2.1. A 2.8 Å Zoliflodacin DNA Cleavage Complex with S. aureus DNA Gyrase

Crystals of zoliflodacin in a complex with a 20-mer DNA homoduplex (20-447T) and *S. aureus* DNA gyrase were grown by a microbatch crystallization method, and a 2.8 Å dataset was collected on beamline I24 at Diamond Light Source (see Materials and Methods for details). The data were phased using the 2.5 Å QPT-1 complex with the same 20-447T DNA and *S. aureus* DNA gyrase in the same P6_1_ space group (PDB code: 5CDM; a = b = 93.9 Å, c = 412.5 Å) and then refined (see Materials and Methods for details and Figure 2 below for electron density). The 20-447T DNA homoduplex contains 18 base-pairs and a G-T mismatch at either end of the DNA.

The structure shows two zoliflodacins binding in the cleaved DNA, physically blocking religation (Figure 3). The DNA has been cleaved by and is covalently attached to tyrosine 123A from the GyrA subunit (and to the symmetry related tyrosine 123A’, from the second GyrA subunit in the complex). Catalytic metal ions (normally Mg^2+^ in bacteria) are required for DNA cleavage, and in our structure we can see two Mn^2+^ ions occupying the ‘B-site’ in the GyrB and GyrB’ subunits [10].

### 2.2. Zoliflodacin Interacts with GyrB, Whereas Moxifloxacin Interacts with GyrA

Figure 4 compares the binding sites of compounds in our 2.8 Å zoliflodacin structure with a 2.95 Å *S. aureus* DNA gyrase DNA cleavage complex with the widely used quinolone antibiotic moxifloxacin (PDB code: 5CDQ [34]). Figure 4a shows the binding mode of zoliflodacin with the pyrimidinetrione (or barbituric acid moiety) of the compound, making direct interactions with GyrB. In particular, the terminal oxygen of the pyrimidinetrione makes a hydrogen bond with the main-chain NH of aspartic acid B437.

This contrasts with moxifloxacin, where the compound (Figure 4b,e) interacts with S84A and E88A from the GyrA subunit via the now well-characterized water–metal ion (Mg^2+^) bridge [12,33,34,35,46,47]. The lack of interactions with GyrA and the interactions with GyrB account for the much of the activity of zoliflodacin against quinolone-resistant strains of bacteria (e.g., Table 4 in [3]; target-mediated resistance is common in quinolone resistant bacteria [11]). The interactions of the quinolones with the GyrA (or ParC) subunit via the flexible water–metal ion bridge may account for some of the specificity of the quinolones for DNA gyrase and topo IV (see sequence alignment in Figure 5) over the two human type IIA topoisomerases, Top2α and Top2β.

However, the residues on GyrB, which partly form the DNA gyrase–zoliflodacin binding interactions, are conserved not only in the bacterial type IIA topoisomerases but also in the human enzymes. As shown in the sequence alignment in Figure 5 and in Figure 4a, zoliflodacin recognizes and interacts with the GD from the conserved EGDSA motif and the RG from the PLRGK (or PLKGK in Gram-negative DNA gyrases). While E435 at the start of the EGDSA motif is a catalytic residue, the other residues from the EGDSA are not catalytic, neither are the PLRGK motif residues. In QPT-1, only the GD and RG residues from GyrB contact the compound.

The specificity of spiropyrimidinetriones, such as zoliflodacin, towards bacterial type IIA topoisomerases such as human topoisomerases was proposed to be because such compounds can be squeezed out of the pocket when the DNA-gate closes in human topoisomerases [34]. An alternative explanation could be because the DNA gate of DNA gyrase acts like a pair of swing doors, closing automatically once the transport segment has been pushed through [34]. This alternative explanation might account for the lower activity seen against both human topo 2s (liabilities) and bacterial topo IVs, which tend to only act on supercoiled DNA (DNA cross-overs).

Conformational flexibility in spiropyrimidinetrione ligands, such as QPT1 and zoliflodacin, may be important in allowing ligands to maintain favorable interactions within the binding sites as the DNA wriggles the protein [34,49,50]. In addition to the multiple tautomeric forms that the pyrimidinetrione moiety can adopt (only one of which is chemically called a ‘pyrimidinetrione’) and the conformational flexibility of the anilino-nitrogen [34], methyl-oxazolidine-2-one may also be able to adopt more than one conformation. Multiple high-resolution structures will be required to fully discern how the compound wriggles (when its binding pocket changes shape) as the enzyme is moved around by its substrate DNA [50]. However, from this initial 2.8 Å zoliflodacin structure, it is clear that the major protein interactions made by zoliflodacin are clearly with the GD and the RG from the highly conserved EGDSA and PLRGK motifs (Figure 4 and Figure 5).

In the 2.1 Å crystal structure of the NBTI GSK299423 with the *S. aureus* gyrase^CORE^ and DNA (PDB code: 2XCS; [13]), a Y123F mutant was used so that the DNA could not be cleaved. In this 2.1 Å GSK299423 structure, the +1:+4 base-pair (Figure 1c) occupies a similar space to the inhibitors in the zoliflodacin and moxifloxacin structures. Some reasons for the conservation of the EGDSA and PLRGK motifs (Figure 5) may be discerned from this 2.1 Å structure. While the side-chain of E435 (the first residue of the EGDSA motif) coordinates the catalytic metal (at the ‘A’ position, poised to cleave the DNA), both the main-chain NH and side-chain hydroxyl of serine 438 are within the hydrogen-bonding distance of the phosphate between nucleotides 1 and 2. The main-chain C = O of Arg 458 and the main-chain NH of Lys 460 (from the PLRGK motif) accept and donate hydrogen bonds to the -1 guanine base, helping to hold it firmly in place. NBTIs can stabilize complexes with one strand cleaved or with no DNA cleavage [13,24,51]; however, experimental nucleotide preferences for NBTI cleavage have not yet, to the best of our knowledge, been determined [52].

### 2.3. Target Mediated Resistance to Zoliflodacin in N. Gonorrhoeae

The binding of zoliflodacin to the conserved motifs on GyrB correlates well with the low prevalence of target-mediated resistance; only one of some 12,493 *N. gonorrhoeae* genomes from the PathogenWatch database has a predicted first-level resistance mutation [53]. Assessing the probability of developing resistance is an important step in the development of any new antibiotic. The development of zoliflodacin (AZD0914) for gonorrhea followed from a 2015 paper assessing the likelihood of developing resistance in *N. gonorrhoeae* [48]. This paper showed that higher MIC resistance was associated with target mutations in three amino acids in *N. gonorrhoeae* DNA gyrase, namely GyrB:D429N, K450T, and S467N [48]. These mutations were identified via the in vitro selection of resistance and can give a four-fold to sixteen-fold increase in the MIC of zoliflodacin [48,54]. Interestingly, these *N. gonorrhoeae* GyrB mutations correspond to D437, R458, and N475 in *S. aureus* DNA gyrase. The D429N mutation is associated with the slower growth of bacteria [55]. All three regions are close to the compound (see Figure 4a). In the D437N mutant (*S. aureus* DNA gyrase), the asparagine side chain may have its NH_2_ group pointing towards the compound (because if the sidechain was in the opposite orientation, the hydrogens on the NH_2_ would clash with hydrogens on proline 56, i.e., the P in PLRGK).

Zoliflodacin has an extra methyl-oxazolidine-2-one ring, which QPT-1 does not possess (Figure 1a,b), and this extra ring makes van der Waals contacts with residues N476 and E477. While there is clear electron density for the both the additional fluorine and the extra ring (which are not in the QPT-1 structure; Figure 6), the 2.8 Å electron density map is not able to clearly define all water structures or totally unambiguously define the orientation of the extra five-membered ring (see Figure 2 and Figure 6). N475 is equivalent to the third mutated residue in *N. gonorrhoeae* GyrB, Ser 467 [48]. The mutation of this residue, which is adjacent to residues contacting the compound, presumably affects their conformations. A similar effect is perhaps seen in the *S. aureus* ParC V67A, found in a strain of *S. aureus* resistant to gepotidacin [56]. In high-resolution *S. aureus* DNA gyrase NBTI crystal structures, three residues (A, G, and M) from the GyrA motif 68-ARIVGDVM-75 are within the van der Waals distance of the compounds [13,22,24]. ParC V67A is the first V in the equivalent *S. aureus* ParC sequence 64-AKTVGDVI-71, i.e., Val 67 is adjacent to an amino acid making direct van der Waals contacts with the compounds.

Most bacteria (including *N. gonorrhoeae*) have two type IIA topoisomerases—DNA gyrase and topo IV. The target-mediated resistance to dual targeting quinolones, which form bactericidal DNA cleavage complexes with both DNA gyrase and topo IV, is only significant after mutations have occurred in both DNA gyrase and topo IV [57]. The observation by Alm et al. [48] of mutations only occurring in *N. gonorrhoeae* DNA gyrase when bacteria were challenged with zoliflodacin (AZD0914) suggests the compound has limited activity against *N. gonorrhoeae* topo IV. While third-generation cephalosporin-resistant, fluoroquinolone-resistant *N. gonorrhoeae* was listed as a high-piority target in 2018 [39], *A. baumanii* was a higher priority target.

### 2.4. Improved Activity of Zoliflodacin against A. baumannii Compared to QPT-1

*A. baumannii* was selected as a WHO critical priority AMR pathogen [39] to test for susceptibility to zoliflodacin. The MIC of zoliflodacin against two carbapenem-resistant outbreak strains of *Acinetobacter baumannii* [58] was determined (see Section 4 for details) as 4 µg/mL (Table 1). The tested outbreak strains of *A. baumannii* (Table 1) possessed imipenem and meropenem MICs in excess of 4 µg/mL, precluding their treatment with these carbapenems [58]. The zoliflodacin MIC of 4 µg/mL suggested that although its activity has been optimized against other Gram-negative bacteria, the potency of zoliflodacin against *A. baumannii* is better than that of QPT-1, from which it was developed (the activity of QPT-1 against *A. baumannii* is from Supplementary Table S4 in the paper by Chan et al. [34]; note that QPT-1 is considerably more active against an efflux knock-out strain, *A. baumannii* BM4454 (ΔadeABC ΔadeIJK) [59]). As expected from previous testing, the analysis of the *S. aureus* reference strains NCTC 12981 showed good zoliflodacin susceptibility (<0.313 µg/mL).

## 3. Discussion

Some Gram-negative bacteria are difficult to kill with antibiotics. Not only do they have two cell walls but they also have export pumps that can rapidly pump antibiotics out of the bacteria [59]. Such bacterial export pumps can play a role in antimicrobial resistance [60]. There is much interest in compounds that can inhibit antibiotic efflux pumps [61], as there is clearly a potential for combination therapies. If the MICs of a compound such as zoliflodacin could be lowered, the dose might be lowered and the therapeutic window would be increased.

However, the success of whole-cell screening, including early counterscreening of human cells for safety, seems to have been effective in discovering two new classes of Gram-negative targeting antibiotics [5,13]. A similar approach, although starting with a natural product, recently lead to the discovery of evybactin, a new *M. tuberculosis* DNA gyrase-targeting compound [62]; this compound appears to work in a similar manner to the thiophene inhibitors [63] that allosterically stabilize DNA cleavage complexes [64] by binding to a ‘third site’—a hinge pocket [10].

Interestingly, two of the mutations in *N. gonorrhoeae* GyrB that give rise to resistance to zoliflodacin (AZD0914) are Asp429Asn and Lys450Thr, which correspond in *S. aureus* crystal structures to residues involved in making up the binding pocket of the compound (Figure 4). Namely, Asp 437 (=Asp429) is the D from the conserved EGDSA motif and Arg 458 (=Lys450) is from the conserved PLRGK motif.

In a previous paper describing the crystals structures of QPT-1, moxifloxacin, and etoposide in DNA cleavage complexes with *S. aureus* DNA gyrase [34], the DNA gate of DNA gyrase was proposed to act like a pair of swing doors, through which the T-segment could be pushed (Figure 1f) but that would then swing close. Such a model might partly account for why in *N. gonorrhoeae* mutations are only seen in GyrB and not in ParE [48]. The swinging close of the DNA gate in DNA gyrase might be predicted to give slower ‘off’ rates for zoliflodacin compared to topo IV; it was also proposed that zoliflodacin would be squeezed out of a slightly larger equivalent pocket in human topo2s [34]. Much work remains to be done; for example, one current model suggests that before the C-gate (or exit gate) can be opened, the small Greek key domain senses the presence of the T-DNA segment (once it has passed through the G-gate) and then moves the catalytic metal away from the active site (see the Supplementary Discussion and Supplementary Figures S12 and S13 in [34]). This model allows the DNA to be religated by the lysine residue from the highly conserved YKGLG motif at the C-terminus of the Greek key domain (see Figure 5), while not allowing DNA cleavage by the catalytic metal when the exit gate is opened and not allowing exit gate opening while the gate DNA is cleaved. In this model, this is a ‘safety feature’ of type IIA topoisomerases, allowing DNA religation by the YKGLG lysine but inhibiting DNA cleavage by the catalytic metal. Interestingly, it has also been shown that DNA gyrase can catalyze supercoiling by introducing a single nick in the DNA [65]; perhaps this mechanism is also a safe way of introducing negative supercoils into DNA without opening the C-gate.

The safety and size of the therapeutic window are clearly important in antibacterial drug discovery. It will be interesting to see if the new spiropyrimidinetrione class of compounds, such as zoliflodacin, can be developed to be safer and more efficacious medicines with less of a tendency for target-mediated antibiotic resistance than the quinolones.

## 4. Materials and Methods

### 4.1. Protein Purification and Crystallization of a Zoliflodacin DNA Cleavage Complex

The *S. aureus* DNA gyrase fusion truncate GyrB27:A56 (GKdel) (M_w_ 78,020) was expressed in *E. coli* and purified based on the procedure used by Bax et al. [13], modified as described [25]. The protein was purified (at 10 mg/mL = 0.128 mM) in 20 mM HEPES pH 7.0, 5 mM MnCl_2_, and 100 mM NaSO_4_. The DNA oligonucleotide used in crystallizations, 20-447T, was custom-ordered from Eurogentec (Seraing, Belgium). Received in lyophilized form, the DNA was resuspended in nuclease-free water and annealed from 86 to 21°C over 45 min to give the duplex DNA at a concentration of 2 mM. The zoliflodacin was purchased from MedChemExpress (South Brunswick Township, NJ, USA) as a solid and was dissolved in 100% DMSO, forming a 100 mM stock solution.

Crystallization complexes were formed by mixing a protein, HEPES buffer, DNA, and compound and incubating the mixture on ice for 1 h 15 min. Crystals of *S. aureus* GyrB27:A56 (GKdel)-zoliflodacin-20-447T were grown using the microbatch under oil method [38], with streak seeding being implemented for subsequent plates after the first plate gave crystals. Following established protocols, a crystallization screen consisting of Bis-Tris buffer pH 6.3 to 6.0 (90, 150 mM) and PEG 5*k*MME (13–7%) was used. For a single drop, 1 µL of complex mixture was mixed with 1 µL of crystallization buffer in a 72-well Terasaki microbatch plate, prior to covering with paraffin oil. The plates were incubated at 20 °C and crystal growth was observed between 5 and 30 days. A seed solution was prepared by crushing several previously grown hexagonal rod-shaped *S. aureus* GyrB27:A56 (GKdel)-zoliflodacin-20-447T crystals in 20 µL of crystallization buffer. The crystal, which gave a 2.8 Å dataset, was grown in a crystallization plate, where 1 µL of complex mixture (0.066 mM GyrB27:A56 dimer, 0.171 mM 20-447T DNA duplex, 5.714 mM zoliflodacin, 2.571 mM MnCl_2_, and 342.9 mM HEPES pH 7.2) was mixed with 1 µL of crystallization buffer (90 mM Bis-Tris pH 6.3, 9% PEG 5 *k*MME). A large single crystal was transferred to a cryobuffer (15% glycerol, 19% PEG 5*k*MME, 1 mM zoliflodacin, 5% DMSO, 81 mM Bis-Tris pH 6.3) before flash-cooling in liquid nitrogen for data collection.

### 4.2. Data Collection, Structural Determination, and Refinement

The data were collected (3600 x 0.l° degree images) on beamline I24 at Diamond Light Source. The data were processed and merged with dials [66,67,68], as shown in Table 2. A low-resolution cutoff of 25 Å was applied when manually reprocessing the data with dials to avoid problems with the backstop shadow. The high-resolution cutoff was determined by having a CC_1/2_ > 0.30 [69]. The structure was refined starting from the 2.5 Å complex with the same DNA and the related compound QPT-1 (PDB code: 5CDM) [34,38]. The data, which are not twinned and are in space group P6_1_, were reindexed (H = k, K = h, L = -l) to be in the same hand and of the same origin as other liganded structures in the same space group (e.g., PDB codes: 2XCS, 4BUL, 5IWI, 5IWM, 5NPP, 6QTK, 6QX1, and 5CDM). The initial rigid body refinement of 5cdm-BA-x.pdb (P6_1_ cell: a = b = 93.88 Å, c = 412.48 Å) reduced the R-factor (R-free) from 0.3900 (0.3899) to 0.2684 (0.2743). Further refinement with phenix.refine [70,71] and refmac [43,72] gave the final structure (Table 3), which had a reasonable geometry. Restraints for zoliflodacin were generated in Acedrg [73]. As we were interested in structures with ligands and inhibitors, we used the standard BA-x numbering scheme throughout [10] (the coordinates, 8bp2-BA-x.pdb, are available from a table of structures from the ‘research’ tab of Ben Bax’s website at Cardiff (https://www.cardiff.ac.uk/people/view/1141625-bax-ben, accessed on 4 January 2023). This means zoliflodacin inhibitors in sites 1 and 1’ have CHAINID I (for the inhibitor) and residue numbers 1 and 201 (see Figure 3 in [10]). In this 2.8 Å zoliflodacin *S. aureus* DNA gyrase structure, the chains are named as B (GyrB) and A (GyrA) from the first fusion truncate subunit and D (GyrB) and C (GyrA) from the second subunit (the BA-x nomenclature stands for GyrB/GyrA extended numbering). The DNA strands have CHAINIDs E and F (see [10] for further details). The electron density maps for the inhibitors are shown in Figure 2. The water structure near the inhibitors was based on that in the 2.5 Å structure with QPT-1 (PDB code: 5CDM). The water and glycerol structures were based on the electron density maps and higher resolution structures (the 1.98 Å *S. aureus* complex PDB code 5IWI, which contains over 940 water molecules, was superposed).

At each DNA cleavage site, a single catalytic Mn^2+^ ion is seen at the B-site [10]. The electron density on His C 391 was interpreted as being due to a Mn^2+^ ion coordinated by a Bis-Tris buffer molecule, which mediates a crystal contact with one end of the DNA. This interpretation of the electron density was confirmed by re-refining the original 2.1 Å structure of GSK299423 with the *S. aureus* gyrase^CORE^ structure [13]; originally this electron density had been misinterpreted as being due to the DNA. The new interpretation explains why both Bis-Tris and Mn^2+^ ions are needed in the crystallization buffer when growing P6_1_ crystals of *S. aureus* gyrase^CORE^ with ligands.

### 4.3. Structural Analysis

The van der Waals contacts with the ligands (defined as 3.8 Å or less) were calculated with ‘contact’ from the CCP4 suite of programs [75]. The structures were superposed using coot [76] or with limited sets of defined Cαs using LsqKab from the CCP4 suite [75].

### 4.4. Minimum Inhibitory Concentration Assay

The MICs of zoliflodacin against two carbapenem-resistant outbreak strains of *A. baumannii* (BCC 807, BCC 810) [58] were determined in triplicate using the modified broth microdilution reference method ISO 20776-1:2019 [77], as recommended by the EUCAST (European Committee on Antimicrobial Susceptibility Testing) [78]. The concentration range tested was between 40 and 0.313 µg/mL in two-fold serial dilutions. *S. aureus* NCTC 12981 was used as a quality control strain, as the MICs for *S. aureus* have previously been reported.

## Figures and Tables

**Figure 1 ijms-24-01634-f001:**
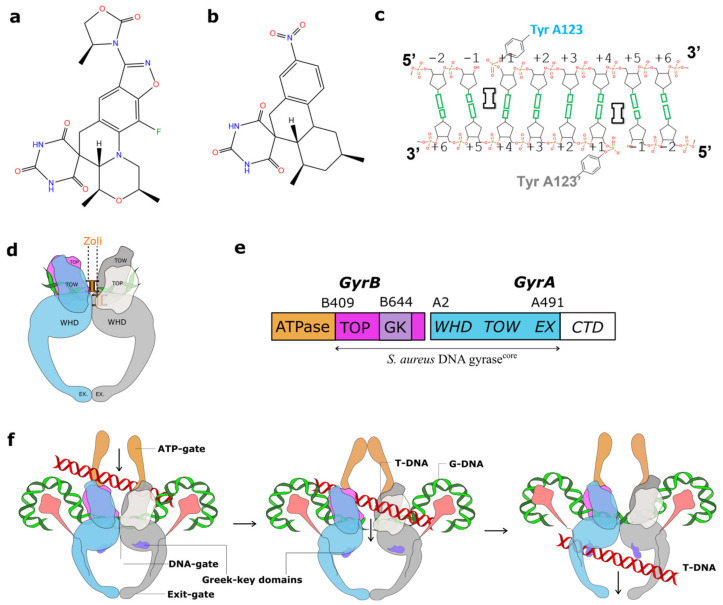
Zoliflodacin, QPT-1, and *Staphylococcus aureus (S. aureus)* DNA gyrase DNA cleavage complexes. (**a**) Chemical structure of zoliflodacin (oxygens shown in red, nitrogens in blue, fluorine in green) and (**b**) chemical structure of QPT-1. (**c**) A schematic of the central eight base-pairs of DNA, with two inhibitors (I) binding in the cleaved DNA and inhibiting DNA religation. Note that DNA cleavage takes place between the −1 and +1 nucleotides on both the Watson and Crick strands. (**d**) A schematic of the DNA cleavage complex with two zoliflodacins of *S. aureus* DNA gyrase and DNA presented in this paper. (**e**) The *S. aureus* DNA gyrase^CORE^ construct consists of residues B409 to B644 from GyrB, fused to A2 to A491 from GyrA. The small Greek key (GK) domain has been deleted from GyrB [13]. (**f**) A simplified schematic of DNA gyrase, in which a G-DNA duplex (green) is cleaved by the enzyme, and another DNA duplex (known as the T or transported DNA, red) is moved through the enzyme. The Greek key domains are not involved in cleaving the gate (or G-) DNA segment [10,13]. The C-terminal domains (CTD) are shown in pink (approximate positions as in full-length *E. coli* structures [14]).

**Figure 2 ijms-24-01634-f002:**
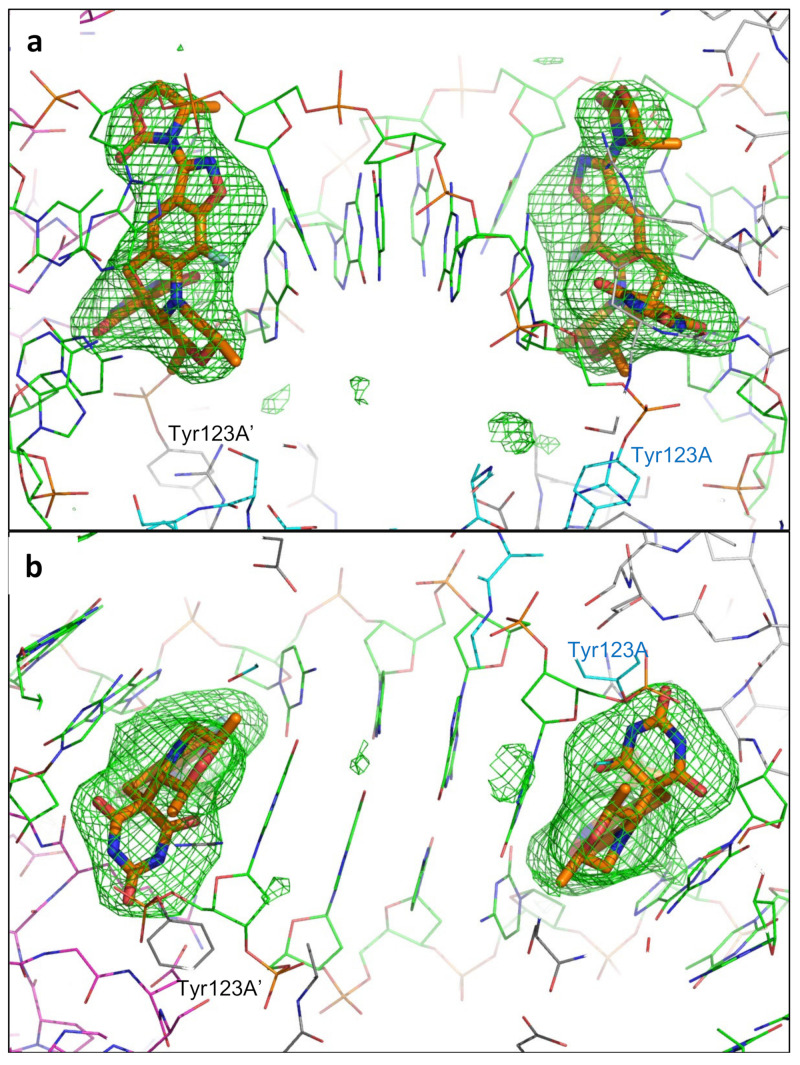
**Final Fo-Fc omit map (+3 sigma) for the two zoliflodacins: orthogonal (90°) views**. A final Fo-Fc omit map was calculated by omitting the two zoliflodacins from the coordinates, which was refined with refmac [43]. The initial omit maps calculated from 5cdm coordinates were of similar quality but showed some additional density around the novel five-membered ring (top (**a**)). The zoliflodacins are shown as sticks with orange carbon atoms; gyrA has cyan or grey carbons, gyrB has magenta or grey carbons, and the DNA has green carbons (water molecules are not shown for clarity). The pyrimidinetrione rings are clearly seen in the view in panel (**b**).

**Figure 3 ijms-24-01634-f003:**
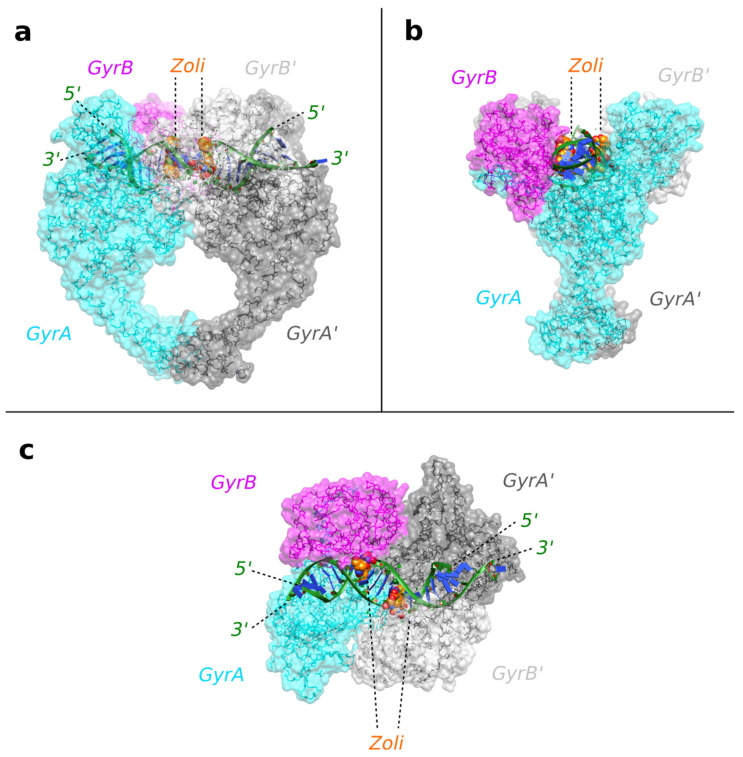
**The 2.8 Å zoliflodacin crystal structure with *S. aureus* DNA gyrase.** (**a**) View of the 2.8 Å zoliflodacin crystal structure. The DNA (cartoon; green backbone and blue bases) has been cleaved by *S. aureus* gyrase (shown as backbone trace with semi-transparent surface). Tyr 123 (and Tyr 123′) have cleaved the DNA and are covalently attached. The compounds (Zoli: zoliflodacin) are shown as solid spheres (carbons as orange, oxygens as red, nitrogens as blue). (**b**) An orthogonal (90°) view of the same complex. (**c**) An orthogonal (90°) view looking down the two-fold axis of the complex. The two ends of the DNA duplex adopt different conformations due to crystal packing. Figure produced using ChimeraX [44,45].

**Figure 4 ijms-24-01634-f004:**
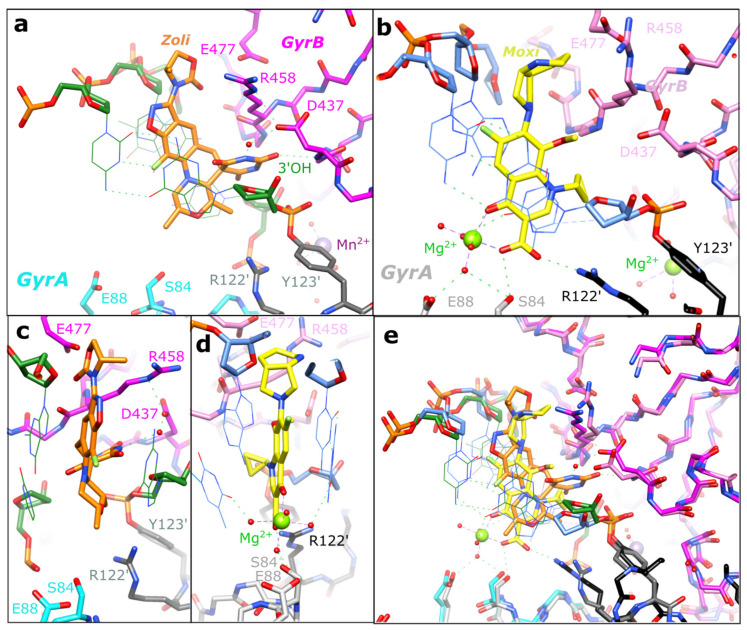
**Equivalent views of zoliflodacin and moxifloxacin in *S. aureus* DNA gyrase DNA cleavage complexes**. (**a**) A close-up view of a zoliflodacin (Zoli: orange carbons) binding site in the 2.8 Å structure. The pyrimidinetrione moiety of zoliflodacin interacts directly and indirectly (via a water) with Asp437 of GyrB (dotted green lines), in a similar manner to that described in the 2.5 Å QPT-1 structure. Y123′ has cleaved the DNA and formed a ‘phosphotyrosine’ type linkage with the cleaved DNA. The DNA-backbone is shown with a fatter ‘stick’ representation, with the bases drawn in thinner ‘line’ (base-pair H-bonds only shown for the +1, +4 base-pair). (**b**) In the 2.95 Å moxifloxacin (Moxi: yellow carbons) structure, the quinolone-bound Mg^2+^ ion (green sphere) and coordinating water molecules (red spheres) make hydrogen bonds (dotted red lines) to S84, E88, and the bases either side of the DNA cleavage site (at the +1 and -1 positions; see panel d). (**c**,**d**) Orthogonal (90°) views of the compound binding sites in the zoliflodacin structure (**c**) and moxifloxacin structure (**d**). (**e**) Superposition of (**a**,**b**). Figure produced using ChimeraX [44,45].

**Figure 5 ijms-24-01634-f005:**
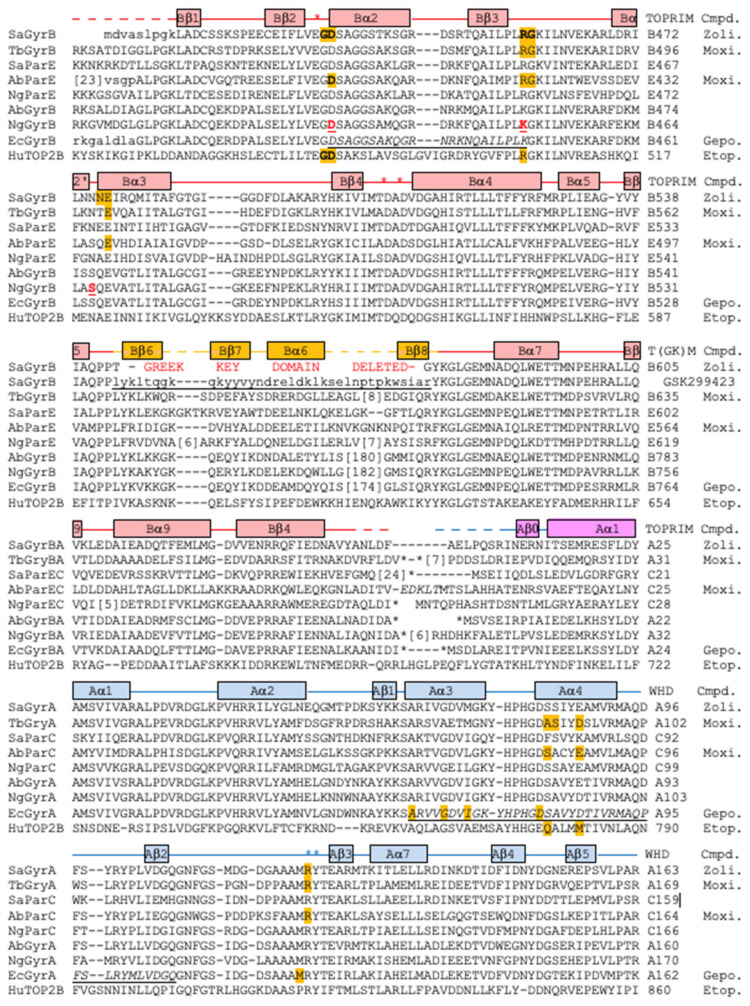
**Sequence alignment highlighting residues contacting zoliflodacin, moxifloxacin, gepotidacin, or etoposide.** An alignment of the residues in the TOPRIM, Greek key, and WHD domains of *S. aureus* GyrB/GyrA (SaGyrBA), *M. tuberculosis* GyrB/GyrA (TbGYRba), *S. aureus* ParE/ParC (SaParEC), *A. baumannii* ParE/ParC (AbParEC), *N. gonorrhoeae* ParE/ParC (NgParEC), *A. baumannii* GyrB/GyrA (AbGyrBA), *N. gonorrhoeae* GyrB/GyrA (NgGyrBA), *E. coli* GyrB/GyrA (EcGyrBA), and human Top2B (HuTOP2B). Numbers in brackets (e.g., [180]) show numbers of residues not included in the alignment; note the large insertion within the Greek key domain in Gram-negative DNA gyrase sequences. The three *N. gonorrhoeae* GyrB residues whose mutations give low levels of resistance are highlighted in red [48]. Amino acids are **highlighted** on sequences if they contact (<3.8 Å) compounds in *S. aureus* DNA gyrase structures with compounds. Zoli. = contacts in the 2.8 Å zoliflodacin structure (pdb code: 8BP2). Moxi. = contacts from the 2.4 Å *M. tuberculosis* complex with moxifloxacin (pdb code: 5bs8) or contacts from the 2.95 Å moxifloxacin complex (pdb code: 5cdq, note for moxifloxacin, the Mg^2+^ ion and water molecules of the water–metal ion bridge are taken as part of the compound; contacts in the 3.25 Å *A. Baumannii* topoIV moxifloxacin complex structure, 2XKK, are nearly identical to those shown). Gepo. = from the 2.37 Å gepotidacin with uncleaved DNA (pdb code: 6qtp; note contacts in the 2.31 Å structure 6qtk and in the full-length *E.coli* cryoEM gepotidacin structures, e.g., pdb code:6rks, are very similar). The contacts mapped onto the HuTOP2B structure are from the 2.16 Å etoposide complex with human Top2β (pdb code: 3QX3; but are similar in *S. aureus* crystal structures: 5cdp and 5cdn). The secondary structural elements in the 3.5 Å *S. aureus* gyrase complex with DNA and GSK299423 (2xcr) are shown above the alignment; note deletion of the Greek key domain gave a GSK299423 structure at 2.1 Å (2xcs). Note: * above the sequence alignment indicates positions of catalytic residues (Glu B435, AspB508, Asp B510, Arg A122, and Tyr A123 in S. aureus DNA gyrase). The quinolone-resistance-determining region (QRDR), defined as 426-447 in *E.coli* GyrB 67-106 in *E.coli* GyrA, is underlined in italics on the EcGyrB/A sequences [11].

**Figure 6 ijms-24-01634-f006:**
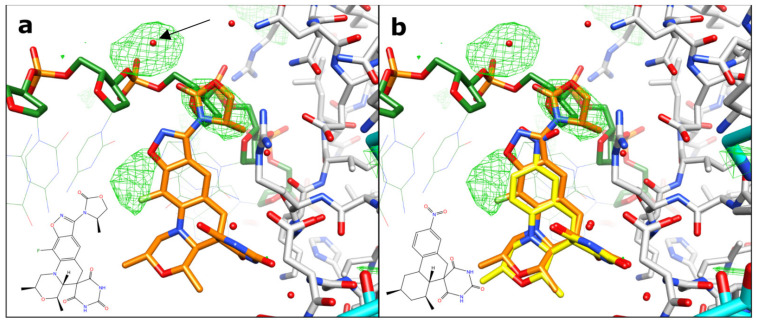
**Difference in map densities from refined QPT-1 structure against zoliflodacin data**. (**a**) The F_o_-F_c_ map from refined QPT-1 coordinates contoured at 3σ shows extra features in the zoliflodacin structure not in the QPT-1 starting coordinates. The arrow points to extra density modelled as water, which could be a metal ion (it is close to the oxygen of a phosphate from the DNA backbone and also an oxygen from methyl-oxazolidine-2-one). (**b**) Refined QPT-1 coordinates are also shown with yellow carbons.

**Table 1 ijms-24-01634-t001:** MICs of zoliflodacin for *A. baumannii*.

Compound	Species	MIC (µg/mL)
Zoliflodacin	*A. baumannii* BCC 807 (UK OXA-23 clone)	4
Zoliflodacin	*A. baumannii* BCC 810 (South East OXA-23 clone)	4
QPT-1 QPT-1	*A. baumannii* BM4454*A. baumannii* BM4454 (ΔadeABC ΔadeIJK) *	16 0.125

* *A. baumannii* BM4454 (adeABC and adeIJK) is equivalent to BM4652 [59].

**Table 2 ijms-24-01634-t002:** Data collection statistics.

	ZOL-2.8
PDB code	8BP2
Diffraction source	I24, DLS
Wavelength (Å)	0.9999
Resolution range (Å)	24.85–2.78 (2.83–2.78) *
Space group	*P*6_1_
Unit cell	94.54, 94.54, 417.13, 90, 90, 120
Total reflections	974,457 (46,716) *
Unique reflections	52,704 (2554) *
Multiplicity	18.5 (18.3) *
Completeness (%)	100.0 (100.0) *
Mean I/sigma(I)	5.9 (0.4) *
Wilson B_factor_	62.4
R_merge_	0.286 (3.872) *
R_meas_	0.294 (3.980) *
R_pim_	0.067 (1.297) *
CC_1/2_	0.997 (0.318) *

* Numbers in brackets are in the outer (2.83–2.78) resolution shell.

**Table 3 ijms-24-01634-t003:** Refinement statistics.

	ZOL-2.8
PDB code	8BP2
Resolution range (Å)	24.85–2.80 (2.87–2.80)
Completeness (%)	99.41 (94.60)
No. of reflections, working set	48,785 (3421)
No. of reflections, test set	2538 (185)
Final Rcryst	0.1957 (0.372)
Final Rfree	0.2375 (0.374)
Cruickshank DPI (Å) *	0.325
No. of non-H atoms (total)	11,713
Protein	10,574
DNA	801
Zoliflodacin	70
Other ligands (Mn, glycerol etc.)	42
Water molecules	226
RMS deviations	
Bonds (Å)	0.009
Angles (°)	1.569
Average B factors (Å2)	
Protein	96.024
DNA	85.103
Zoliflodacin	84.491
Other ligands (Mn, glycerol etc.)	88.911
Waters **	75.396
Ramachandran plot	
Favored regions	96%
Additionally allowed	4%
Outliers	0%

* The Cruickshank DPI (Å) was calculated using the Online_DPI server [74] (Kumar et al., 2015). ** Water molecules were placed where there were water molecules in higher resolution structures (e.g., 5CDM and 5IWI).

## Data Availability

The 2.8 Å zoliflodacin structure has been deposited with the protein databank (the PDB) with the code 8BP2. The PDB nomenclature for compounds and metal ions is inconsistent between different structures. The sensibly named PDB files, following the BA-x nomenclature [10], are available from a table of structures from the ‘research’ tab on Ben Bax’s website (https://www.cardiff.ac.uk/people/view/1141625-bax-ben, accessed on 4 January 2023).

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
