# Peer review of "A 2.8 Å Structure of Zoliflodacin in a DNA Cleavage Complex with Staphylococcus aureus DNA Gyrase"

_ijms, 2023, doi:10.3390/ijms24021634_

Round 1

Reviewer 1 Report

Morgan et al describe the X-ray crystal structure of DNA gyrase in complex with zoliflodacin. The structure shows zoliflodacin interacts with GyrB but does not make interactions with GyrA. Further, residues that may be associated with resistance are close. Understanding these interactions and the binding mode are key to elucidating the mechanism of action of zoliflodacin.

The manuscript is well written and easy to follow with clear figures. I recommend its publication with only a few minor edits:

The omit map for the ligand (currently Supplemental Figure 1) should be in the main text. There should also be a closer view of the density/ligand as its quite hard to discern the quality of the map from the current figure. The method for omit map calculation should also be detailed in the Method or the Figure legend.

A brief explanation for why A. baumannii was chosen for the MIC experiments, and not other Gram-negative bacteria like E. coli or Klebesiella, needs to be at the beginning of section 2.5. The MIC for the two strains against (example) carbapenems should also be stated somewhere in the text for reference. Do the authors believe Zoliflodacin is similarly affected by efflux like QPT-1? The CLSI breakpoint for A. bauminnii should also be mentioned here.

Line 119. This should read the data were phased using the 2.5A QPT-1 complex… rather than the structure was solved.

Author Response

Reviewer: The omit map for the ligand (currently Supplemental Figure 1) should be in the main text.

Answer: Done. We have now included a new version of the omit map as Figure 2 in the main paper.

Reviewer: There should also be a closer view of the density/ligand as its quite hard to discern the quality of the map from the current figure.

Answer: Done. The new omit map shows detailed orthogonal views of both zoliflodacins.

Reviewer:  The method for omit map calculation should also be detailed in the Method or the Figure legend.

Answer: Done. The method for the new omit map calculation is included in the new figure legend.

Reviewer: A brief explanation for why A. baumannii was chosen for the MIC experiments, and not other Gram-negative bacteria like E. coli or Klebesiella, needs to be at the beginning of section 2.5

Answer: Done. We have re-written the beginning of section 2.5 - to explain this in detail. We note that A. baumannii was investigated as a WHO critical priority pathogen. Original testing of QPT-1 by Chan et al. 2015 had examined A. baumannii, P. aeruginosa, E. coli and K. pneumoniae, but we selected to proceed with follow up of zoliflodacin against A. baumannii as the prime example of a critical AMR priority pathogen. In addition, the expense of obtaining zoliflodacin precluded extensive testing of multiple species.

Reviewer:  The MIC for the two strains against (example) carbapenems should also be stated somewhere in the text for reference.

Answer: Done. See “The outbreak strains of A. baumannii tested (Table 1) possessed imipenem and meropenem MIC in excess of 4 µg/mL precluding their treatment with these carbapenems [57].”

Reviewer: Do the authors believe Zoliflodacin is similarly affected by efflux like QPT-1?

Answer: Yes we believe so. We have not yet substantiated that zoliflodacin is efflux by Gram-negative bacteria, but testing in the presence of PABA is currently underway.

Reviewer: The CLSI breakpoint for A. bauminnii should also be mentioned here.

Answer: Since zoliflodacin is currently still within clinical trials and not yet being used under standard clinical practice guidelines, CLSI breakpoints have not been derived. We can only report an in vitro MIC for the compound at the current time.

Reviewer: Line 119. This should read the data were phased using the 2.5A QPT-1 complex… rather than the structure was solved.

Answer: Done. Now line 134 ' The data were phased using the 2.5 Å QPT-1 complex ..

Reviewer 2 Report

This is a nice and well-conducted work, which sheds light into the binding mode of zoliflodacin to the DNA gyrase from Staphylococcus aureus. The work is generally well written, and the results are clearly presented. Is the opinion of this referee that this manuscript might inspire several drug design studies as well as is potentially able to trigger microbiology investigations on the mechanism of action of new generation antibacterials. 

A number of minor concerns have been found:

1)Abbreviations: authors use some common abbreviations that have not been declared in the text (e.g., topo IV; zoli; etc.)

2)Antibacterial properties of the compound should be better described:

a- why A. baumannii has been selected as the target pathogen? A comment on this point is needed, because structural studies were conducted on the protein from S. aureus and one would expect to read on the antibacterial effect against S. aureus (at least). Why this pathogen was not investigated?

b- sequence alignment between DNA gyrase from S. aureus and A. baumannii shall be shown and discussed (e.g. is it expected that the binding mode will be approximately the same?) to support biological activity data.

c- are the MIC tests performed in triplicate? If yes, what is the SD?

d- what is the MIC against each of the two drug-resistant strains tested in this work?

Author Response

A number of minor concerns have been found:

Reviewer: 1)Abbreviations: authors use some common abbreviations that have not been declared in the text (e.g., topo IV; zoli; etc.).

Answer: Done. Topoisomerase IV (topo IV) is now declared on first mention. Zoliflodacin is only abbreviated as Zoli in some figures and is defined explicitly in figure legends. 

Reviewer: 2)Antibacterial properties of the compound should be better described:

a- why A. baumannii has been selected as the target pathogen? A comment on this point is needed,

Done. We have have justified that A. baumannii was selected as a WHO critical priority pathogen as per the same request from reviewer 1.

because structural studies were conducted on the protein from S. aureus and one would expect to read on the antibacterial effect against S. aureus (at least). Why this pathogen was not investigated?

Done. We did investigate the MIC of zoliflodacin against S. aureus but did not report them. We have now stated in the text that zoliflodacin had good activity as expected against S. aureus.

' As expected from previous testing, analysis of the S. aureus reference strains NCTC 12981 showed good zoliflodacin susceptibility (< 0.313 µg/mL).'

b- sequence alignment between DNA gyrase from S. aureus and A. baumannii shall be shown and discussed (e.g. is it expected that the binding mode will be approximately the same?) to support biological activity data.

Done: The sequence alignment figure now includes both S.aureus and A. baumanii DNA gyrase and topo Iv sequences.

c- are the MIC tests performed in triplicate? If yes, what is the SD?

Yes the MIC tests were performed in triplicate and this is now explicitly stated in the Materials and Methods ' The MIC of zoliflodacin against two carbapenem resistant outbreak strains of A. bau-mannii (BCC 807, BCC 810) [59] were determined in triplicate using the modified broth microdilution reference method ISO 20776-1:2019 [78] as recommended by the EUCAST (European Committee on Antimicrobial Susceptibility Testing) [79].'

However, because the resulting MICs were the same on each test we have not reported an SD.

d- what is the MIC against each of the two drug-resistant strains tested in this work?

The MICs of zoliflodacin against each of the two drug resistant strains is 4 (see Table 1).